# Research on Optimization of Healthcare Waste Management System Based on Green Governance Principle in the COVID-19 Pandemic

**DOI:** 10.3390/ijerph18105316

**Published:** 2021-05-17

**Authors:** Ziyuan Liu, Tianle Liu, Xingdong Liu, Aijing Wei, Xiaoxue Wang, Ying Yin, You Li

**Affiliations:** School of Economics and Management, China University of Geosciences, Wuhan 430074, China; liuziyuan@cug.edu.cn (Z.L.); liuxingdong@bit.edu.cn (X.L.); angie_weii@163.com (A.W.); A1002693596@163.com (X.W.); yinying@cug.edu.cn (Y.Y.); liyou@cug.edu.cn (Y.L.)

**Keywords:** COVID-19 pandemic, healthcare waste, green governance principle, temporary office, route optimization

## Abstract

At present, strategies for controlling the COVID-19 pandemic have made significant and strategic strides; however, and the large quantities of healthcare treatment waste have become another important “battlefield”. For example, in Wuhan, the production rate of healthcare waste in hospitals, communities, temporary storage, and other units was much faster than the disposal rate during the COVID-19 pandemic. Improving the efficiency of healthcare waste transfer and treatment has become an important task for government health and environmental protection departments at all levels. Based on the situation of healthcare waste disposal in Wuhan during the critical period of the pandemic, this paper analyzes and studies green governance principles and summarizes the problems that exist in the current healthcare waste management system. Through the establishment of temporary storage facilities along transit routes, digital simulation and bionic experiments were carried out in the Hongshan District of Wuhan to improve the efficiency of healthcare waste transfer. Furthermore, this study discusses the coordination and cooperation of government, hospitals, communities, and other departments in the healthcare waste disposal process and provides guiding suggestions for healthcare waste disposal nationwide in order to deal with potential risks and provide effective references in all regions.

## 1. Introduction

Since the outbreak of the COVID-19 pandemic, a total number of 48 designated hospitals, 16 mobile cabin hospitals, and 3567 communities in the city have come under the grip and become important links in the joint prevention and control of the COVID-19 pandemic. China joined the International Decade for Natural Disaster Reduction in 1990 and began emphasizing “comprehensive disaster reduction” for the first time, and the construction of medical waste management system officially began after SARS in 2003. At present, medical waste management in China is mainly based on the Regulations on Medical Waste Management [1] and the National Hazardous Waste and Medical Waste Disposal Facilities Construction Plan [2]. The existing medical waste management system has outstanding problems such as lagging efficiency and weakness of the regulatory system at all levels in emergent public health events, and all of this makes it more difficult to meet the demand for medical waste disposal during epidemics.

At present, the exploration of medical waste management indicated by an increasing number of researchers and scholars consists of mostly theoretical and political recommendations. Some scholars have tried to process some data in order to practice the aforementioned management. However, in general, the trial failed due to the lack of research on practical application methods. Timely, safe, and proper disposal of medical waste is the second line of defense for epidemic prevention and control in the event of an unexpected public health event [3]. The combination of a solid theoretical foundation and relevant management system is needed to achieve the optimization of key links and improve the management system efficiently.

Since 2016, Wuhan has implemented an intelligent transportation system for medical waste in the city’s tertiary hospitals to enable intelligent data-based management of medical waste [4]. However, the involvement of the relevant hospitals has not fully occurred, and the relevant system operation performance during the pandemic has not been satisfactory. The lack of coordination between hospitals and related departments has resulted in the timely and effective removal and disposal of medical waste in some hospitals failing. Based on the theory of green governance, this paper first selected a temporary storage place through big data analysis and geographic information system application and used an algorithm to optimize the transport path and perfect the management system to transport as much medical waste as possible in the shortest time. Implementation of the algorithm derived, and an effective transport system will rely on the cooperation of the government, hospitals, communities, and other main bodies to solve the problem of medical waste disposal more effectively and would provide management optimization suggestions for the whole country.

## 2. Literature Review

### 2.1. Green Governance Principles

The green governance principles emerged out of green thoughts and movements in Western countries. From the perspective of the participation of non-governmental subjects, green governance originated from the idea of “green politics” triggered by “green political parties” in the 1970s. Since the 1990s, governments have begun to pay more attention to the problem of ecological and environmental protection and have put forward the concept of “green administration” in an attempt to establish “green government” to combat increasingly severe and urgent ecological and environmental problems, where all parts of society join in efforts to build a “green society” and eventually achieve “green governance” [5].

The concept of the ecological environment has been further developed since the “concept of sustainable development” was proposed at the United Nations General Assembly in 1987. The status of the natural environment has been raised, and the concepts of green economy and green growth have emerged [6]. In order to better balance the relationship between nature and human society and to promote harmony between the two, people put forward the principles of green governance.

Scholars at home and abroad have different views on the connotation of green governance. While research abroad began earlier, scholars focused on environmental management, environmental management systems, and the importance of sustainable development concepts for businesses without explicitly addressing the concept of green governance and related measures. As the emphasis on environmental issues has increased, research has become more specialized. While some scholars argue that environmental governance is a comprehensive and systematic mechanism for improving the environment and enhancing corporate performance for governance purposes [7], Pane focused on organizational learning and practice, arguing that green governance is the process of achieving sustainability, waste reduction, social responsibility, and competitive advantage, which require organizations to integrate organizational goals and strategies through continuous learning and developing innovations [8].

Domestic scholars have studied green governance in-depth and from different perspectives. Yuan Lin argues that the synergistic system of green governance is formed through the synergistic interaction between government green governance, social green governance, and market green governance [9]. Li Wei-an believes that the core of green governance is an effective governance mechanism, which is essentially a “public affairs activity” aimed at building an ecological civilization and achieving green and sustainable development [10]. Some scholars argued from the perspective of “green” derivation that green governance is guided by the concept of green values in order to achieve the goal of “economic-political-cultural-social-ecological” five-in-one, sustainable and harmonious development so as to achieve common governance of public affairs based on the sharing of resources among multiple subjects [11]. As environmental issues have an increasingly significant impact on national politics, economics, culture, etc., the demand for the application of green governance principles in countries is gradually growing. This paper argues that green governance is a practical process by which multiple actors work together to govern public affairs to achieve sustainable development goals. This kind of cooperation includes both internal and external aspects, in which the internal cooperation mechanism means to improve the operational efficiency of the system and realize the sharing of resources between different objects and departments under the same subject and while the external cooperation system means to build an efficient management system by clarifying the division of functions of each subject.

### 2.2. Clinical Waste Treatment

Medical waste contains harmful chemicals and pathogenic microorganisms, which have risk characteristics of infection, such as acute infection and latent infection, so its harmfulness is hundreds or even thousands of times of ordinary household garbage. Because medical waste contains a lot of organic ingredients, it rots easily and begins to smell. If it is handled or transported carelessly, it will lead to the spread of germs and even become the source of disease epidemics [12]. Medical waste disposal is closely related to the ecological environment, disease prevention and control, science and technology, and people’s health and life, so it is the focus of environmental supervision in various countries and regions across the world [13]. With the enhancement of people’s awareness of environmental and self-protection, medical waste disposal has aroused wide concern in society. Most countries have established laws and regulations on the management of medical waste; for example, China has promulgated the Regulations on The Management of Medical Waste and Measures for the Management of Medical Waste in Medical and Health Institutions [14]. Various measures have been taken to manage medical waste, but during the COVID-19 pandemic, the speed, quality, and management capacity of medical waste are in urgent need of significant improvement. Therefore, it is necessary to establish a management system and mechanism covering all aspects of medical waste source control, collection, transportation, treatment, and disposal, so as to ensure safe, efficient, orderly, and scientific treatment of medical waste [15]. The internationally recognized and accepted medical waste management principles are “hierarchical priority management principles”, that is, the first is to reduce the generation of waste, the second is to reuse as much as possible (i.e., to reuse after high-temperature treatment), the third is to promote recycling, and the last is to incinerate with heat recovery. At the same time, it also follows the principles of “polluters pay”, “human health first”, and “prevention first” [16]. The aim of medical waste management is to reduce the pollution of medical waste, the core of which is scientific management and the result of which is human health and a clean environment. Fangyuan [17] pointed out the principles of medical waste disposal in China: reduction, resource recovery, harmless treatment, whole-process management, centralized treatment, and introduction of market competition mechanism.

There are many kinds of medical waste treatment technologies, mainly including incineration, sanitary landfill, high-temperature steam sterilization, chemical disinfection, microwave sterilization, plasma, pyrolysis, and other technologies, but the Chinese mainly adopt incineration, high-temperature steam sterilization, and chemical disinfection technology. The technologies of medical waste treatment have their own advantages and disadvantages. Before sanitary landfill treatment, medical waste needs strict pretreatment, so the investment is huge. Moreover, the impact of the sanitary landfill of medical waste on future generations cannot be predicted. The incineration of medical waste also requires a high cost of disposal and will also cause waste gas pollution, which is more harmful, but many developing countries have adopted this method. The result of pressure steam sterilization treatment is safe, effective, and relatively low in cost. However, this method is not suitable for the treatment of pathological waste, and the treatment effect of drugs and chemical waste is not significant [18]. Microwave sterilization technology has the characteristics of good disinfection effects, energy saving, environmental protection, and convenient operation, but it is not suitable for medical waste that contains hazardous chemical substances.

Although the laws and regulations on medical waste treatment are relatively complete, the treatment principles are very practical, and the treatment technologies are diverse, the problems and difficulties in medical waste treatment emerge one after another. For example, administrative supervision is weak, and the administrative punishment standard is too low, the related legal system is not comprehensive, there is a black market for reselling medical waste, a severe reflux social phenomenon, and the cost to hospitals is high, there are unreasonable charges, lax registration, lax packaging, and lax transportation, etc. These problems have been magnified during the COVID-19 pandemic, so further improvements are urgently needed in medical waste treatment. At present, China’s economy has entered a critical period of transformation, and it needs to further promote intensive, knowledge-based, and ecological development. Ecological construction plays a pivotal role in national and social development. Therefore, green governance and coordinated governance are of vital importance. Based on the theory of green governance, this paper analyzes how to implement green governance by using the medical waste treatment model and multi-agent cooperation, taking the disposal of COVID-19 related medical waste in Wuhan as an example.

### 2.3. Implications of Green Governance Principle for the COVID-19 Pandemic

The outbreak of COVID-19 in Wuhan at the end of 2019 was a typical public health disaster, which was characterized by high contagiousness, rapid spread, and a wide reach. The large amount of medical waste generated daily by hospitals is also a source of infection, becoming a factor that is highly dangerous, and the disposal of this medical waste and the reduction in its threat has become a top priority [19]. The green governance principles have two implications for the treatment of medical waste in epidemics.

Firstly, green governance principles facilitate synergies in multiple sectors, including government, health care institutions, communities, and disposal centers. The process of medical waste disposal is systematic and complex, requiring the cooperation of multiple subjects to accomplish it better. The green governance principle allows the subjects to effectively unite and clarify their respective tasks and work, thus relieving the storage pressure on hospitals and reducing subsequent infections of medical staff.

Secondly, the green governance principles are conducive to local, sustainable development. The accumulation of medical waste in the open not only increases the risk of infection for the medical personnel handling it but also poses a potential danger to the air, soil, and other natural resources, and green governance will avoid such problems, eradicate them from the root causes and promote the recovery of the economy and ecology after the epidemic.

## 3. A Model for Medical Waste Disposal Based on Green Governance Principle

### 3.1. Disposal of Medical Waste during Epidemic Period in Wuhan

In 2014, the Wuhan Institute of Environmental Science had pointed out that the surface antigen rate of hepatitis B in medical waste was 89%, the total number of bacteria was 8.9 × 1010/g, and the coliform count was 8.3 × 1014/g, with a high risk of infection [20]. According to a study, the SARS-COV-2 virus has strong survivability and can survive on surface materials (such as, metal, plastic, and glass) for up to 9 days [21], so it is necessary to attach great importance to the timely disposal of medical waste to reduce the possibility of infection. During the critical period of epidemic prevention and control in Wuhan, the average daily output of medical waste fluctuated around 100 tons. The disposal capacity of medical waste has now increased from 50.5 tons/day to 262.8 tons/day. The disposal capacity of medical waste has been greatly improved. However, there are still the following problems and shortcomings that need to be solved urgently.

First, the production base of medical waste is large, and the problem of transshipment and disposal is lagging behind. Starting from March 2020, the medical waste accumulated in Wuhan has been completely cleared and basically cleaned out on a daily basis, but the daily collection of medical waste is still in a state of rapid increase. The limited temporary storage capacity of the disposal center leads to the lag of medical waste transportation, the open air and overflowing accumulation of highly infectious medical waste in the community and hospitals, and the fact that medical waste is highly contaminated, presenting a significant risk of cross-infection to logistics staff.

Second, the centralized disposal site is scattered, and the transfer time cost is high. At the beginning of the pandemic, there were only two operating units qualified for medical waste disposal in Wuhan, but the production rate of medical waste far exceeded the capacity of treatment at that time. Although the mobile and emergency “mobile medical waste incineration tank” was slowly established in the later stage, many weak links were exposed. It is difficult for medical waste transport vehicles from all regions of the city to transport medical waste to the nearest disposal point with disposal allowance quickly.

Third, the process of medical waste disposal is cumbersome, and human resources are insufficient. It is difficult to unify the frequency and time of medical waste transfer and need artificial registration information signature confirmation. At present, the treatment of medical waste is mainly realized by manual operation of the logistics department and community staff. Lack of human resources, large per capita operation, and low accuracy have become the most prominent problems in medical waste management.

### 3.2. Medical Waste Treatment

#### 3.2.1. Clinical Waste Management System

The problems in the management of medical waste disposal in China under the emergent public health event have created an urgent need to explore new medical waste management systems. To this end, we have developed a medical waste management model based on green governance principles, which emphasize multi-discipline coordination and resource integration of various departments during an emergent public health event. The model advocates for the establishment of a coordinated external management system supervised by government health and environmental protection departments, hospitals designated by the city, and qualified incineration disposal units. In the model, third-party logistics companies and third-party logistics companies cooperate with each other to realize information sharing and separation of power and responsibility. The model also uses a rational allocation of material and human resources among the relevant departments within each institution and a system of dedicated personnel to achieve internal coordination in the transport of medical waste. Overall, the model structure of medical waste management under green management was formed to improve the linkage efficiency of the medical waste management system and to build a green administration by the government. Green production by hospitals, communities, green storage in temporary storage, and a green disposal center with a multi-body and collaborative green management system that uses big data and informatization are also included, as shown in the model presented in Figure 1.

In order to simplify the workflow and reduce the error rate of manual processing, by establishing a data-sharing platform and using information technology, the tracking, management of the entire process of medical waste from generation to disposal is realized, and all information is updated in real-time in a shared database. Government departments can view the latest data in the back office at any time to monitor and record the whole process of sound disposal, improving the efficiency of the linkage. Third-party logistics should be integrated with the fourth-party logistics supply chain during transit, with all transport vehicles equipped with GPS positioning systems. As a result, all drivers can scan QR codes to obtain medical waste capacity information and use data transmission calculations before transit to ensure adequate transport capacity.

Medical waste is identified and sealed by the logistics department after it has been generated in the community. In designated hospitals and in the medical cabin, the medical waste is labeled and stored, and the relevant data are entered into the information platform of the medical waste transit staging facility. Third-party logistics companies are the first method to dispatch GPS-located medical waste carloads when they receive the latest medical waste generation information and check the latest capacity information registered by drivers to select vehicles that have sufficient spare capacity and are closest to the source of generation. Drivers of medical waste carriers are required to register up-to-date information on the capacity of the vehicle and the starting point of the journey by scanning the QR code of the medical facility, transiting drop-off point and disposal center before and after the loading and unloading of medical waste, At the same time, hospitals should register the transport time and volume of medical waste with an appointed person’s real name, the medical waste collected by the community should be registered by the leader of the community workstation to reduce the burden on hospitals and communities, reducing the secondary cross-infection caused by the virus in areas with high population density. After receiving the notice that the disposal center is about to have the disposal allowance, the nearest transport vehicle will rush to the large storage area to complete the transfer of medical waste.

#### 3.2.2. Temporary Storage Sites Selection

In order to relieve the pressure of medical waste storage in hospitals and to reduce the risk of cross-contamination of residents in the community due to the accumulation of medical waste, there is a need for timely medical transfer of waste masks, protective clothing, etc., generated in the community to sites with sterilization conditions. To alleviate the pressure of medical waste storage in hospitals and communities, by setting up a temporary medical waste storage place in Wuhan, and realize the three-level dynamic treatment of collection point-transfer temporary storage place-centralized disposal point. The location of the temporary storage should be strictly in accordance with the Regulations on Medical Waste Management and Notice on the issuance of technical principles for environmental impact assessment of construction projects of hazardous waste and medical waste disposal facilities (i.e., for Trial Implementation). These two notices take the social environment, natural environment, site environment, engineering geology, climate, emergency rescue into account, and other factors should try to meet the following requirements:(1)Comply with the specific requirements of Wuhan’s overall development plan and environmental protection professional plant, and make long-term plans;(2)The situation of medical waste generation in the community, the existing disposal capacity of medical waste in Wuhan, the service area for setting up a medical waste transit staging area, transport, land use status, and infrastructure, etc., considering the necessity and feasibility of project construction [22];(3)Sites for the temporary storage of medical waste should be located as far away as possible from schools, densely populated areas, and domestic waste storage sites, and not in the year-round upwind direction of residential areas, taking into full consideration public opinion and suggestions.

To meet the above conditions, we first inquired about the addresses of all hospitals and communities in Wuhan one by one and then established the map function between longitude and latitude and the two-dimensional Cartesian coordinate system. A classical k-means algorithm was introduced to determine the number and location of temporary processing points so as to obtain preliminary results. However, it was difficult to directly add the above three constraints based on green governance in the iteration process because the constraints were difficult to quantify directly. Therefore, we considered the algorithm location results and established a buffer zone of 4 square kilometers (which can be regarded as a fault-tolerant zone). The medical treatment points could be moved to any point in the buffer zone to ensure all constraints.

The distribution of hospitals, communities, temporary storage, and community collection points through GIS is shown in Figure 2. Hospitals represented by serial numbers in Figure 2 are shown in Table 1. It should be noted that the hospital, the community, and the community waste collection point were all existing locations, while the temporary storage point is the location proposed by the K-means algorithm combined with the geographical terrain and the green governance principles.

#### 3.2.3. Systematic Transport of Medical Waste for Transit and Temporary Storage

In emergent public health events, especially during major infectious disease outbreaks, even though medical waste is the same, hospitals and community medical waste have significant differences in characteristics. Specifically, the wide distribution and relatively small amount of community medical waste require community workers to collect medical waste from various communities and workstations and take it to medical waste collection points for unified treatment. However, the distribution of medical waste in hospitals is relatively concentrated and dense. To ensure the full coverage of the model in Wuhan, we divided the medical waste into two transport routes before the temporary storage is transported, as shown in Figure 3. The medical waste of the hospital is transported directly to temporary storage, and the community medical waste is collected centrally through the community collection point and then transported to the nearest temporary storage. There are two main lines of the medical waste transport network; one is from the community collection site to temporary storage and disposal sites, the other one is from hospital to temporary storage to disposal site, medical waste is orderly transported to the disposal site for disposal.

The path optimization of the medical waste management model based on green governance principles can distinguish between hospitals and community transport routes according to the optimization results during vehicle dispatch; thereby reducing the risk of infectious disease pathogen spread in hospitals and providing reference advice for setting up community collection points in different regions, reducing the retention time of medical waste in the community, reducing the risk of secondary infection spread by pathogens through medical waste in the community, and promoting the timely disposal of medical waste.

### 3.3. Medical Waste Transport Route Optimization and Simulation

#### 3.3.1. Path Optimization Algorithm Analysis

In this paper, vehicle path planning was applied to the process of medical waste transfer and treatment in hospitals and communities [23]. At present, the study of the path optimization problem adopts various algorithms. In this section, we introduce three commonly used algorithms: the genetic algorithm (GA), ant colony algorithm (ACO), and the simulated annealing algorithm (SA), and analyze their advantages and disadvantages. Finally, from the perspective of applicability and efficiency, we propose a novel hybrid model GA-ACO, which combines GA and ACO algorithms to achieve the medical waste transportation optimization problem studied in this paper.

GA is a computational model that simulates the natural selection process of Darwin’s biological evolution by encoding the parameters into chromosomes. GA simulates the behaviors of genetic mutation and crossover of chromosomes and finally achieves global optimization through screening. That is, the optimal path is the optimal individual after the survival of the fittest.

ACO is an optimization algorithm that simulates ant foraging behavior. It was first proposed by Italian scholar Dorigo M and others in 1991 [24] and was first used to solve the TSP (Traveling Salesman Problem). The ACO uses the concept of pheromone to simulate the communication mechanism of individual mutual communication, giving artificial ants a certain memory capacity. Therefore, the selected path will gradually become optimized as the number of beats increases [25].

SA is a probabilistic method to obtain the globally optimal value of a function. Specifically, it is a meta-heuristic approximation global optimization problem in a large search space, which constantly jumps out of the locally optimal value through “annealing”. SA is usually used in problem scenarios where the search space is discrete (for example, the traveling salesman problem). SA has been used in the path planning of material distribution of Yonghui supermarket in Xiamen city. When the algorithm was solved, the path solution was shortened from the initial random solution of 131.29 km to the optimal solution of 71.294 km, which greatly reduced the logistics distribution cost [26]. Although each of these heuristic algorithms has advantages, there are certain limitations. By consulting the relevant information, we compared the three algorithms from the perspective of global convergence, efficiency, and applicability [27]. The comparison is shown in Table 2.

Through the previous comparison of the three heuristic algorithms, in view of the high global convergence and applicability of the GA, and easy integration with other algorithms, the GA was used to solve this problem. By comparing the previous three heuristic algorithms, it can be seen that the GA has high global convergence and applicability but low efficiency [28]. Considering it is easy to combine with other algorithms and the ACO algorithm has a better convergence speed, efficiency, and applicability, a hybrid algorithm combining GA and ACO was established in this paper to optimize the medical waste transportation path by taking advantage of the convergence speed and solving efficiency of both.

#### 3.3.2. GA-ACO Model in Medical Waste Transport

The model simulation scenario in this paper was in the Hongshan district, Wuhan. Community location data was derived from the Open Street Map Wuhan land use map, and the community population data was derived from the Wuhan population density map. Using a clustering algorithm on all communities in the Hongshan district, we found 28 community medical treatment points for waste collection (hereinafter referred to as community collection points). We chose seven temporary waste storage locations referring to the Standard for Pollution Control on Medical Waste Treatment and Disposal [29].

The method for combining GA and ACO was as follows. ACO was taken as the basic framework, introducing the genetic and mutation ideas in GA into the pheromone of ACO so that the pheromone had attenuation and mutation behavior. The purpose of the combination was to prevent the algorithm solution from falling into the local optimal value. The purpose of this study was to use the shortest path to transport the most medical waste from community collection points and hospitals to temporary storage locations. The overall goal of model optimization needed to consider the three factors of transport distance, amount of transported medical waste, and amount of remaining medical waste. To better describe the operation process of the algorithm, the relevant elements are expressed as follows.

Twenty-eight community collection points, eight hospitals, and seven temporary storage locations were sequentially identified with integers from 0 to 42. $P_{m}=[P_{i_{1},j_{1}},P_{i_{2},j_{2}},\ldots ,P_{i_{n},j_{n}}]$ represents a complete medical waste transport route of the garbage truck with the number m passing through n sections of paths. R is the total amount of medical waste on the day, $P_{m}^{c}$ is the total amount of waste collected by the garbage truck numbered m on the corresponding route P. $D_{ij}$ is the distance between point i and point j, λ is the hyper-parameter.

Because there was only one garbage disposal point in Wuhan, the model removed the path optimization from the temporary storage to the garbage disposal point. The assumptions regarding the capacity of medical waste transport vehicles, the amount of hospital medical waste, and the amount of community medical waste were all based on the average level in Wuhan. 

Model assumptions:(1)The transport vehicle departs from a temporary storage location, passes through several community collection points and hospitals, and can eventually return to any temporary storage location;(2)The temporary storage location can accommodate the total amount of garbage transported by the transport vehicle;(3)The distance between the starting point and the ending point of the path is calculated based on the longitude and latitude of the Baidu coordinate system, without considering complex terrain and other factors;(4)During the epidemic, the community produced 5 kg of medical waste per 1000 people per day, the transport capacity of the garbage truck was 4.5 tons, the community collection point collected medical waste generated by the community within 4 square kilometers. All of the above data came from official announcements.

Based on the above assumptions, the optimization goal of medical waste transportation is expressed as follows:(1)$$\max{\frac{\lambda_{1}\sum_{m}P_{m}^{c}}{\lambda_{2}\sum_{m}\sum_{P_{i,j}\in P_{m}}D_{ij}+\;\lambda_{3}(R-\sum_{m}P_{m}^{c})}}_{}$$
S.T.
(2)$$\sum_{m}P_{m}^{c}<=R$$
$$\mathrm{where}\;P_{m}=[P_{i_{1},j_{1}},P_{i_{2},j_{2}},\ldots ,P_{i_{n},j_{n}}]$$
(3)$$j_{1}=i_{2},j_{2}=i_{3},\ldots ,j_{n-1}=j_{n}$$
(4)$$i_{1},j_{n}\in [35,42].\;j_{1}\in [0,35]$$

Among them, Formula (1) is the optimization goal. The total amount of medical waste transported in a day is positively correlated with the optimization goal, and the overall transport distance is negatively correlated with the optimization goal. Formula (2) guarantees the basic inequality relationship between the total amount of medical waste delivered and the total amount of waste on the day. Formula (3) guarantees the continuity of the various paths in the route, and Formula (4) guarantees that the transport vehicle must start from the temporary storage location, pass hospitals or community collection points, and eventually return to any temporary storage location. Overall, the essence of the combination of GA and ACO was to combine the concept of pheromone in ACO and the idea of gene mutation in GA.

In our study, the pheromone represents the probability that vehicles from each temporary storage location will go through this point to the rest of the locations. Mathematically expressed as a probability vector:(5)$$I=[I_{1}^{m_{1}},I_{1}^{m_{2}},\ldots ,I_{1}^{m_{n}},\ldots ,I_{7}^{m_{1}},I_{7}^{m_{2}},\ldots ,I_{7}^{m_{n}}]$$
where $I_{1}^{m_{n}}$ represents the probability that the freight car from the first temporary storage will go to point $m_{n}$ via this place, and n represents the number of possible choices for the next waypoint. The initial setting had equal probability. In the practice of algorithms, pheromones could be divided into global pheromones and local pheromones. Before each simulation experiment, the local pheromone copies the global pheromone. When the amount of rubbish at a certain route point is emptied during operation, the probability of going to that point in the local pheromone is set to 0, thereby sharing the information with other transport vehicles, and the global pheromone was used for updating after each iteration.

After establishing the pheromone, all the community collection points, hospitals, and temporary storage places were used as the path points of ACO. Each waypoint contained multiple dimension information: longitude, dimension, medical waste volume, and pheromone.

When doing each iteration, we updated the pheromone as follows:(6)$$\tau_{ij}(t+n)=(1-\rho )\tau_{ij}(t)+\;\Delta \tau_{ij}(t)$$

Formula (6) shows the pheromone update on path (i,j) at time (t+n), $\tau_{ij}(t)$ represents the pheromone at path (i,j) at time t, and ρ represents the information volatilization factor. After each iteration, a relatively short path was selected in this iteration to complete the pheromone update, thereby increasing the probability of selecting the path at the same intersection next time. At the same time, pheromones do not simply increase, but also decrease with time as I=0.85I, in order to prevent invalid paths from being repeatedly explored. In the hybrid model, this article used the core ideas of coding, inheritance, and mutation.

Inspired by the GA algorithm, a random factor = 0.7 was introduced to prevent the model from falling into the local optimal or repeatedly exploring the wrong path under the interference of pheromone. This means that each time a path was selected, it had a 70% probability of being selected according to the pheromone of the path point (gene inheritance) and a 30% probability of random selection (gene mutation).

After around 100 iterations, we could find the shortest path according to the pheromone in each path nodes. The shortest path here was not the shortest spatial distance but abstracted into the optimization target in Formula (1). The flow chart of the algorithm is shown in Figure 4.

In order to further verify the superiority of the hybrid algorithm, this paper compared it with four other algorithms. The objective function was set as the amount of medical waste (unit: kg) transported per unit distance (unit: km). After several times of simulation experiments, the optimal solution could be obtained through the hybrid algorithm (GA-ACO). The experimental results are shown in Table 3.

#### 3.3.3. Simulation of Medical Waste Path Optimization through Simulation Experiments

The experiment scene temporarily took the Hongshan District of Wuhan as the simulation scene of the medical waste transport route optimization model. The GA-ACO model was used to solve the transport path simulation, the partial path planning diagram of Hongshan District is shown in Figure 5.

The establishment of temporary storage could take about 28 tons of medical waste in Hongshan District to be settled reasonably every day, thus avoiding the pollution caused by improper treatment. During the simulation experiment, the core idea of the optimization goal is to transport more medical waste from hospitals and community collection points to temporary storage in the shortest time through fewer paths. After a large number of experiments and comparisons, the number of temporary storage locations was seven. The location was set near Luoyu Road 965, Shendun first Road, and other places in Hongshan District. The obtained optimal path planning result was 0.1 kg of medical waste per 1 km path.

#### 3.3.4. Promote Efficient Operation Optimization Management in Simulation Experimental Scenarios

During the disposal of medical waste, the functional departments must implement a double-link system, and the medical waste information generated by the COVID-19 pandemic link should be filled out separately and submitted to the medical waste management information platform on a daily basis by hospitals. The infectious medical waste generated could be distinguished from the medical waste link generated in other clean areas so as to facilitate ladder collection, transport and priority collection, and disposal. Additionally, it could promote the orderly and efficient transfer and disposal of medical waste, as far as possible, through the means of information and computer efficient data processing, and to effectively cooperate with the heads of the various functional departments to promote the orderly and efficient transfer of medical waste.

In order to ensure the effective cooperation of all departments during the epidemic period, the efficiency of vehicle transport could be reasonably arranged, the disposal process could be tracked and managed, and an information platform was established for the information exchange of government departments supervision, medical waste information of hospitals, community medical waste information, transit logistics information and capacity information data of temporary storage, and the visualization of information and auxiliary resource scheduling through an intelligent city-wide big data network as well as opening information could be realized.

## 4. Recommendations for the Application of Green Synergistic Management of Medical Waste

The government should establish special supervision and management of the second-level ladder management, composed of medical waste collection points. The government also needs to take the health and environmental protection departments of the Hubei Provincial Government as the leading core. The sanitation and environmental protection departments of the Wuhan and district governments are the centralized disposal institutions and temporary warehouses for the direct supervision departments (Figure 6).

In regards to the working mechanisms of the above-mentioned bodies, the following recommendations are made to promote the implementation of a hierarchical medical waste management system.

### 4.1. Standardized Management of Effective Ladder Construction at Community Concentration Points and Temporary Storages

(1) All closed communities should adopt a management system that places a single person in charge of each building. In principle, one appointed person is allocated to every 20 households, and the sealed-packed medical waste generated by the residents of the building should be collected and centralized every 2 days and transported to the community medical waste-centering sites.

(2) During an epidemic, medical waste storage sites should be established as soon as possible. Masks worn by suspected and confirmed patients and other highly infectious medical wastes generated during medical observation and treatment should be separated and sealed in two packages and two-colored in accordance with relevant national norms to reduce the risk of transmission. The head of each community should rely on the community health service center, in principle, to set up a medical waste collection site in each region and be managed by a dedicated person who would be responsible for collecting medical waste collected and generated by the nearby community medical waste bins, community service stations, personal clinics, and community health centers and timely registration of the amount and time of transfer through the data exchange platform before transfer in order to achieve digital mobile phone management.

(3) Municipal and district people’s governments should be responsible for the implementation of a special person in charge of the nearby community medical waste bins, community service stations, individual clinics, and community health centers to collect and generate medical waste to the community medical waste centralized temporary storage sites in an effective route.

(4) Medical waste staging facilities should be equipped with professional personnel to carry out classification, marking, and sterilization work in strict accordance with technical specifications to avoid the growth of viruses and bacteria in confined spaces. At the same time, medical waste in temporary storage should be transferred and disposed of by disposal enterprises, and the transfer of medical waste to unqualified enterprises should be strictly prohibited.

### 4.2. Effective Interdepartmental Oversight of the Whole Process

(1) Strengthen supervision. Supervision of the supervision of medical waste centralized disposal enterprise voucher operations should be strengthened. For enterprises engaged in the centralized disposal of medical waste, they must hold a hazardous waste operating permit issued by the relevant government departments. Enterprises that have not obtained a hazardous waste operating permit must not engage in the operation of centralized disposal of medical waste. Secondly, departments such as the Health and Planning Commission, the Ministry of Ecology and Environment, and the Ministry of Transport, in accordance with their division of responsibilities, should conduct regular inspections of the collection, temporary storage, transport, and disposal of medical waste in all regions, carry out risk assessments, and order rectification within a limited period of time for the existence of potential environmental risks.

(2) Increase financial security. Timely approval of personnel and vehicles for transport, personnel, vehicles, and work requirements should be adhered to, and the implementation of paid services for medical waste disposal should be put in place. The specific cost standards should be determined by the Municipal Price Bureau in conjunction with the Municipal Health and Planning Commission in accordance with the current costs of medical waste collection, transfer, and disposal by the number of beds or weight. Hospitals and communities ought to adhere strictly to the standards. At the same time, a combination of paid services and regional subsidies for disposal enterprises should be implemented to ensure full coverage, collection, and disposal.

(3) Strengthen departmental coordination and establishing information-sharing mechanisms to realize the exchange of key information during special periods. Dynamic management of the city’s hospitals, transport units, and disposal enterprises should be established through the development of the corresponding transport and management system for the list and changes in medical waste collection, temporary storage, transport, disposal, and other information. National Health Commission of the People’s Republic of China, the Ministry of Ecology and the Environment, the Ministry of Transport, the Ministry of Public Security, the Development and Reform Commission, and other departments should strengthen their collaboration to ensure, on the basis of the implementation of industry regulation making sure that the priority is given to the collection and disposal of epidemic-related medical waste. In respect to joint law enforcement, joint inspections and joint cases should be regularly carried out to crack down on medical waste-related violations, establish and improve the work assessment mechanism and accountability system, and provide special management departments to supervise the disposal of medical waste from collection to arrival at the disposal center, to avoid the illegal sale of medical waste, and to monitor the whole process of medical waste disposal.

### 4.3. Emergency Management Systems in Place for Medical Facilities and Medical Facilities in the Mobile Cabin

(1) To increase the internal publicity and guidance of hospitals, departments broadcast the hospitals “new coronavirus pneumonia prevention and control knowledge” and “the use of medical protective equipment” training videos should be made to popularize medical waste classification knowledge to medical personnel, patients, and cleaning staff so that everyone could be educated and knows the relevant prevention and control knowledge. A team of garbage sorting supervisors and volunteers should be established, composed of community residents who are in good physical condition to strictly enforce the management system and guide the hospital staff and people who seek medical treatment to consciously sort and put in household garbage. Medical waste should be sealed with double red special packaging and prohibited to be opened after sealing. In particular, the injurious medical waste should be included in the infectious medical waste management, together with the sharps box, before packaging, and transported by a dedicated vehicle with a dedicated person in charge. Each hospital should have implemented separate registration and management of medical wastes related to the epidemic, and filled in the transfer slip separately from other medical wastes, strictly implemented the transfer slip system, made good records, and established corresponding collection, transport, storage, and disposal accounts.

(2) The information based on medical waste management in designated hospitals in Wuhan should be established and improved to achieve full coverage and dead-end management of medical waste in institutions and to shorten the time interval for declaring the type, volume, flow, storage, and disposal of medical waste to environmental protection departments during the outbreak of unexpected public health events, so as to facilitate the adjustment of subsequent disposal work.

(3) The monitoring of the hospital’s medical waste transient storage during special periods should be strengthed, specifically to check whether the location, area, equipment, drainage, and vector biological control and disinfection measures are reasonable, including whether the cryogenic storage equipment could be used normally (including the energization) or whether antiseptic treatment requires a separately sealed package.

## 5. Future Research Directions

Firstly, the specific application strategies and path optimizations for each algorithm in the green governance principle model comparison presented in this paper have the potential for further in-depth study. In particular, in the application of the effective combination of hybrid algorithms and big data, the effective combination of different algorithms through data processing, regionalization, intelligence, and management system will trigger further enrichment and refinement of the ladder management model through comparative experimental research. Secondly, regarding the validation of the model building outcome pathway optimization, this paper only used the green governance principles for pathway optimization of the process of medical waste transfer treatment in primary care institutions and communities. However, what other factors may affect the efficiency of management operations in business, government, hospitals, and other aspects? For example, in optimizing the route in densely populated areas, how to improve the contingency plan for environmental emergencies such as transport links in the face of constraints and how to prevent the impact of environmental risks during transport in transport time or traffic peaks, and how to establish a complete and effective collaborative medical waste disposal system are all very interesting topics and research directions.

## 6. Conclusions

In summary, this paper argues that the application of green governance principles to the study of the optimal path for community transient storage and the optimal choice of stepped path for disposal is effective under certain conditions. Firstly, there is a need to focus on the multiple principles of instrumentality, controllability, and effectiveness in the operational selection of green governance principles. Secondly, in terms of achieving optimal outcomes, it is important to follow the principles of goal management identity, scientific algorithm use, and convenience to ensure that the optimal acquisition can be achieved in practical applications through simulation and experimental path optimization under green governance model construction. Finally, on the path of intelligent development of science and technology applications, attention should be paid to enhancing the effectiveness of the application and practice of integrated management of diversity in order to make effective, safe, and rational use of resources.

## Figures and Tables

**Figure 1 ijerph-18-05316-f001:**
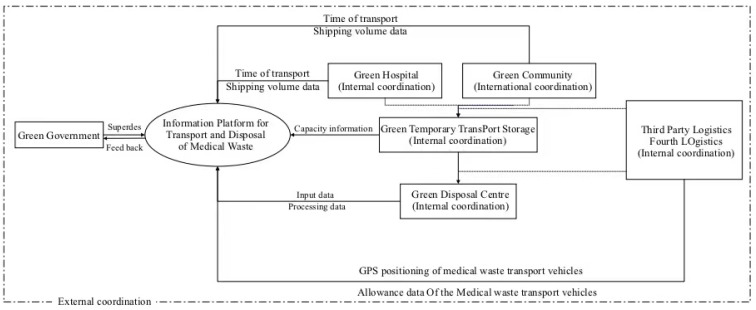
Medical waste management model based on green governance principles.

**Figure 2 ijerph-18-05316-f002:**
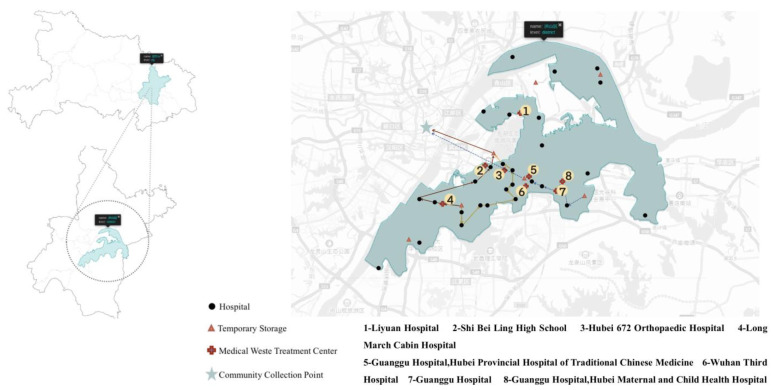
Distribution of health facilities, communities, temporary storage, community collection points.

**Figure 3 ijerph-18-05316-f003:**
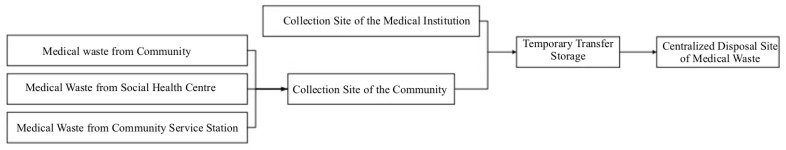
Medical waste transport routes.

**Figure 4 ijerph-18-05316-f004:**
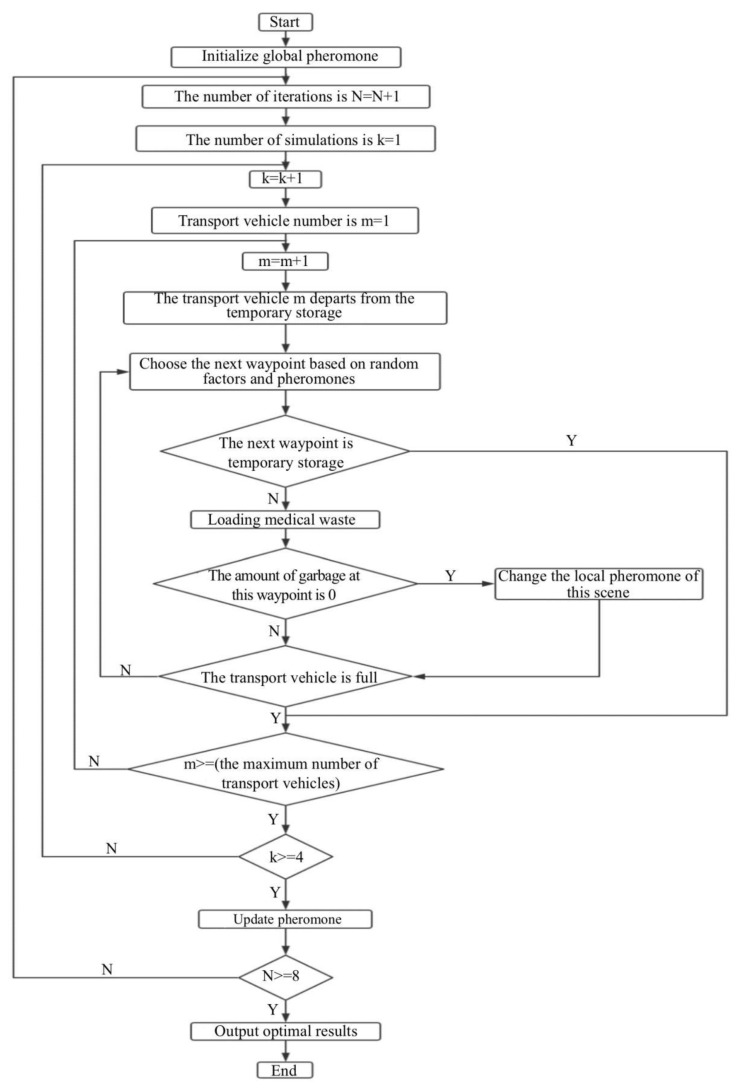
The flow chart of the algorithm.

**Figure 5 ijerph-18-05316-f005:**
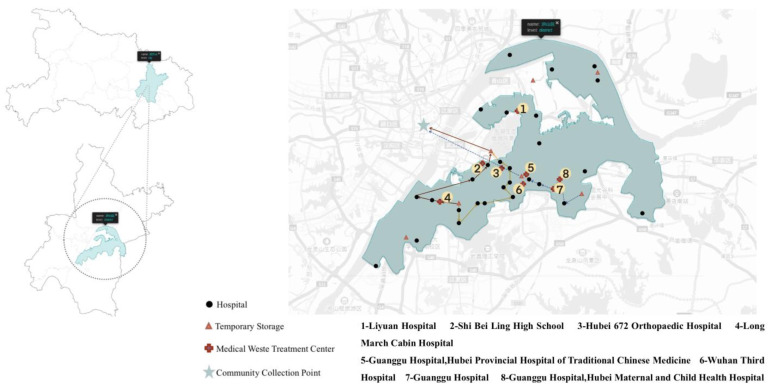
Partial interception in Hongshan District based on the path planning of temporary storage system.

**Figure 6 ijerph-18-05316-f006:**
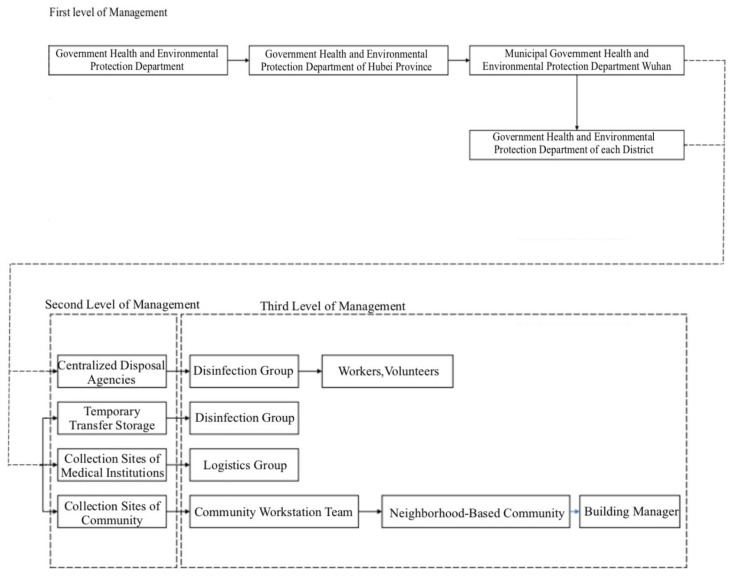
Medical waste hierarchy management system.

**Table 1 ijerph-18-05316-t001:** Hospitals represented by serial numbers in Figure 2.

Number	Name
1	Liyuan Hospital
2	Shi Bei Ling High School
3	Hubei 672 Orthopaedic Hospital
4	Long March Cabin Hospital
5	Guanggu Hospital, Hubei Provincial Hospital of TCM
6	Wuhan Third Hospital
7	Guanggu Hospital
8	Guanggu Hospital, Hubei Maternal and Child Health Hospital

**Table 2 ijerph-18-05316-t002:** Comparison of three algorithms in global convergence, efficiency and applicability.

Algorithm	Global Convergence	Efficiency	Applicability
GA	High; it can get the globally optimal solution	Low; it has low convergence efficiency, especially in the later period may “premature” to further reduce the convergence efficiency	High; suitable for most models and many problems that cannot be built
SA	Low; the result is likely to be non-globally optimal	High	Low; because it is easy to achieve the local optimum, so it is generally not used alone
ACO	Higher than SA; it is possible to get a global optimum	Higher than GA; early lack of “pheromone” leads to slow convergence	Higher than GA; it has strong applicability to graphic problems, and some problems can be transformed into graphic problems

**Table 3 ijerph-18-05316-t003:** Comparison of objective function values of optimal solutions of various algorithms.

Algorithm Name	Optimal Solution Objective Function Value kg/km
GA	1.14
SA	0.97
ACO	1.23
GA-ACO	1.56

## Data Availability

https://pages.semanticscholar.org/coronavirus-research (accessed on 1 March 2021). GitHub—microsoft/COVID-19-Widget: Terms and reference files for COVID-19 Widget.

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
