# Peer review of "Research on Optimization of Healthcare Waste Management System Based on Green Governance Principle in the COVID-19 Pandemic"

_ijerph, 2021, doi:10.3390/ijerph18105316_

Round 1

Reviewer 1 Report

This paper tried to build a healthcare waste management system, based on the concept of green governance, using algorithm to optimize the transport path. This research topic is interesting and covers several important practical issues. It is expected that the study may point out feasible solutions to the emerging problems of healthcare waste management, especially in the period of the pandemic such as COVID-19.

The authors may consider to address the following issues:

  1. It seems that the phrase “Green Governance Theory” is not commonly used in academic works; instead of, “green governance principles”, “green governance perspectives”, “green governance initiative”, “green governance innovation”, “green governance mode”, etc. are more widely applied. Please double check.
  2. The name of correspondence is not in the list of authors’ names.
  3. The authors should consider to improve the arrangements and statements in this manuscript. It seems that several parts were not well-organized and the main ideas were not indicated clearly. Besides, the redundant information also can be found in those parts; the lengthy and clumsy sentences and paragraphs make the audiences feel confused and unreadable. For example, is there any relationship between “green governance theory” and “collaborative governance theory”? Why the information of “collaborative governance theory” was introduced in “Literature review”? Is that information useful for the research paper? If the introduction of two kind of theories (“green governance theory” and “collaborative governance theory”) is for a comparison purpose, then why the former one was selected to be applied in the research. And in Section 3.3.1 (page 10), three algorithms (namely ACO, SA, and TS) were introduced. However, the comparison of GA, SA, and ACO was made! What is GA? There is no information of GA which is mentioned in this section (it can only be found later, in page 13, line 540-544). Why was TS included in this section? In addition, “Ant colony algorithm” (AG) was repeatedly mentioned in page 11 (from line 530 to line 539). What is the difference between latter “Ant colony algorithm” (AG) and former “Ant colony algorithm” (ACO) (which was presented in page 10, line 457-459).
  4. Please check the detailed instructions on in-text citation (how to cite the reference sources) and follow exactly guidelines for preparing tables (formatting instructions how to present tables). The authors should check and fix all in-text citations in the manuscript. In terms of the format of the table, header information in the top row and/ or in the first column should be added (check Table 1)
  5. Please enlarge the content in figures (Figure 1, Figure 2, and Figure 6)
  6. Line 397-399 (page 9), Figure 2 was mentioned first, and after that Table 1 was introduced. Therefore, it seems that Figure 2 and Table 1 should also be placed in their proper order (Sequential order).
  7. Figure 2 and Figure 3 should be translocated?
  8. The reason why Hongshan district was selected as an experimental area should be explained.
  9. Please pay more attention when using abbreviation, for example, “osm” (page 11, line 487) should be presented in the full form (Open Street Map) in the first use.
  10. Line 720, page 19, what is “The Wei and Planning Commission”?

Author Response

Dear Editors and Reviewers:

Thank you for your letter and for the reviewers’ comments concerning our manuscript entitled “Research on Optimization of Healthcare Waste Management System Based on Green Governance Theory in COVID-19” (ID: ijerph-1148168 ).

Those comments are all valuable and very helpful for revising and improving our paper, as well as the important guiding significance to our researches. We have studied comments carefully and have made correction which we hope meet with approval. The main corrections in the paper and the responds to the reviewer’s comments are as flowing:

1.It seems that the phrase “Green Governance Theory” is not commonly used in academic works; instead of, “green governance principles”, “green governance perspectives”, “green governance initiative”, “green governance innovation”, “green governance mode”, etc. are more widely applied. Please double check.

Response: We agree with your comment and replace “Green Governance Theory” with “Green Governance Principle”

2.The name of correspondence is not in the list of author’s name.

Response: We are very sorry for our incorrect writing. When submitting the manuscript, the corresponding author made a typo. We will submit an “Authorship Change Form” for correction.

3.The authors should consider to improve the arrangements and statements in this manuscript. It seems that several parts were not well-organized and the main ideas were not indicated clearly. Besides, the redundant information also can be found in those parts; the lengthy and clumsy sentences and paragraphs make the audiences feel confused and unreadable. For example, is there any relationship between “green governance theory” and “collaborative governance theory”? Why the information of “collaborative governance theory” was introduced in “Literature review”? Is that information useful for the research paper? If the introduction of two kind of theories (“green governance theory” and “collaborative governance theory”) is for a comparison purpose, then why the former one was selected to be applied in the research. And in Section 3.3.1 (page 10), three algorithms (namely ACO, SA, and TS) were introduced.

Response: We agree with you that collaborative governance theory is redundant.Therefore, the information about collaborative governance theory is removed from this paper. We've added an introduction to the GA algorithm, and the introduction of TS algorithm is deleted. And we are very sorry that our typo has resulted in an abbreviation error on page 13 of ACO.

4.Please check the detailed instructions on in-text citation (how to cite the reference sources) and follow exactly guidelines for preparing tables (formatting instructions how to present tables). The authors should check and fix all in-text citations in the manuscript. In terms of the format of the table, header information in the top row and/ or in the first column should be added (check Table 1)

Response: We have check and fix all in-text citations and table formats in the manuscript.

5.Please enlarge the content in figures (Figure 1, Figure 2, and Figure 6)

Response: It is really true as Reviewer suggested that content in figures should be enlarged. And we have enlarge the content in figures.

6.Line 397-399 (page 9), Figure 2 was mentioned first, and after that Table 1 was introduced. Therefore, it seems that Figure 2 and Table 1 should also be placed in their proper order (Sequential order).

Response: It was an oversight on our part that Figures 2 and 3 need to be interchanged. And Table 1 should follow Figure 2

7.Figure 2 and Figure 3 should be translocated?

Response: It was an oversight on our part that Figures 2 and 3 need to be interchanged.

8.The reason why Hongshan district was selected as an experimental area should be explained.

Response: Hongshan District of Wuhan is one of the main areas for the outbreak and treatment of COVID-19 in 2020, and Hongshan District is a representative choice.

9.Please pay more attention when using abbreviation, for example, “osm” (page 11, line 487) should be presented in the full form (Open Street Map) in the first use.

Response: We apologize for using the abbreviation Open Street Map. And now it has been changed.

10.Line 720, page 19, what is “The Wei and Planning Commission”?

Response: We changed it to the National Health Commission of the People’s Republic of China.

We would like to express our great appreciation to you and reviewers for comments on our paper. Looking forward to hearing from you.
Thank you and best regards.
Yours sincerely,

Xiaoxue Wang

Corresponding author:
Name: Tianle Liu
E-mail: [email protected]

Reviewer 2 Report

The paper is an overall decent paper that espouses green overnance theory in the waste management realm. The authors gave a decent account of the theories underpinning waste management. Some interesting algorithms have been mooted to enhance waste management in the respective research area. The introduction, literature and methodology are sound but the paper could use some more language and grammar editing  before publication.

Author Response

Dear Editors and Reviewers:

Thank you for your letter and for the reviewers’ comments concerning our manuscript entitled “Research on Optimization of Healthcare Waste Management System Based on Green Governance Theory in COVID-19” (ID: ijerph-1148168 ).

Those comments are all valuable and very helpful for revising and improving our paper, as well as the important guiding significance to our researches. We have studied comments carefully and have made correction which we hope meet with approval. And we sorted out the logic and language of the article, and made careful modifications according to the opinions of the other three reviewers.

We would like to express our great appreciation to you and reviewers for comments on our paper. Looking forward to hearing from you.
Thank you and best regards.
Yours sincerely,

Xiaoxue Wang
Corresponding author:
Name: Tianle Liu
E-mail: [email protected]

Reviewer 3 Report

The authors studied the healthcare waste management system through green governance theory in this manuscript, however, there are some comments for the authors to consider: 

  1. There's no clear research question, nor research significance(s).
  2. In discussing the green governance theory or environmental governance theory, the authors have not provided extensive study. The green/environmental theory can be included other policy actors, such as corporations, NGOs, etc.
  3. The collaboration theory as shown in the manuscript is duplicated with green / environmental theory. This part is suggested to omit. 
  4. And the green governance theory is suggested to apply to the case in this manuscript. 
  5. The method of data collection deployed in this manuscript is unclear. 

The manuscript is strongly suggested to rewrite for consideration. 

Author Response

Dear Editors and Reviewers:

Thank you for your letter and for the reviewers’ comments concerning our manuscript entitled “Research on Optimization of Healthcare Waste Management System Based on Green Governance Theory in COVID-19” (ID: ijerph-1148168 ).

Those comments are all valuable and very helpful for revising and improving our paper, as well as the important guiding significance to our researches. We have studied comments carefully and have made correction which we hope meet with approval. The main corrections in the paper and the responds to the reviewer’s comments are as flowing:

1.There's no clear research question, nor research significance(s).

Response: This paper summarized the disposal problems of medical waste in Wuhan during the epidemic period. Combined with the green governance theory, and proposed and optimized the disposal plan of medical waste. We think it can provided guiding suggestions for the disposal of medical waste in China, so as to deal with potential risks and provide effective reference for all regions.

2.In discussing the green governance theory or environmental governance theory, the authors have not provided extensive study. The green/environmental theory can be included other policy actors, such as corporations, NGOs, etc.

Response: In a major public health emergency, we must solve a matter according to the government's policy. The relevant companies or NGOs are all policy practitioners. They just follow the instructions. Therefore, we do not think it is necessary to discuss the management of these organizations in major public health emergencies. They do what the government tells them to do. Because we all trust the Chinese government.

3.The collaboration theory as shown in the manuscript is duplicated with green / environmental theory. This part is suggested to omit.

Response: Thank you for your advice. We think the collaborative governance theory does not conform to the main content of the article. We have deleted the content about collaborative governance theory.

4.And the green governance theory is suggested to apply to the case in this manuscript.

Response: We developed a medical waste management model based on the theory of the green management, emphasis on public health emergencies during the multi-agent collaboration with various departments of resources integration, set up a regulation by the government, health, environmental protection department, the city's medical institutions, the medical establishment that decide a dot, square treatment qualified incineration disposal unit, the third party logistics and fourth party logistics company to cooperate with each other, achieve information sharing in the process of transition, accrual distinct external collaborative management system.At the same time, the relevant departments within each organization should reasonably allocate material and human resources, and implement a system of dedicated personnel to achieve internal coordination in the process of medical waste transportation.Finally, to improve the linkage efficiency of the medical waste management system, a medical waste management structure is constructed under the green governance of the government, the green production of medical institutions and communities, the green storage of temporary storage in transit, and the green disposal of the disposal center, which integrates multi-agents, collaboration, information technology and big data.

5.The method of data collection deployed in this manuscript is unclear.

Response: This parper of the data is based on the news during the COVID-19 and data from the Ministry of Ecology and Environment of the PRC.

We would like to express our great appreciation to you and reviewers for comments on our paper. Looking forward to hearing from you.
Thank you and best regards.
Yours sincerely,

Xiaoxue Wang

Corresponding author:
Name: Tianle Liu
E-mail: [email protected]

Reviewer 4 Report

Overall Comments:

I wish that I could write an overall summary of what this paper is about but it is very unclear to me what the actual objective is. It seems to be testing various different algorithms under a “green” governance paradigm to solve the medical waste disposal problem in Wuhan. However at one point, it is mentioned that the purpose of the study is to use the shortest path to transport the most medical waste… yet, at another point it says “Based on the theory of green governance, this paper first selects the temporary storage place through big data analysis and geographic information system application, takes the shortest transport time as the premise, uses the algorithm to optimize the transport path and perfect the management system. Furtherly rely on the cooperation of the government, hospitals, communities and other main bodies to achieve the goal of solving the disposal of medical waste more effectively, and providing management optimization suggestions for the whole country, and promote the smooth development of the national economic and ecological recovery work after the epidemic.” Alternatively, the last paragraph in section 2.3 states: “Based on the theory of green governance and collaborative governance, this paper analyzes how to implement the green governance by using the medical waste treatment model and multi-agent cooperation, taking the disposal of the new pneumonia medical waste in Wuhan as an example.” Therefore, it seems like there are multiple objectives and they are not clearly set-up for and described in a well-understood manner. It is not clear how green governance theory actually relates or integrates with the model optimization aspects of this paper. Finally, the first line of the last paragraph states: “this paper argues that the application of green governance theory to the study of the optimal path for community transient storage and the optimal choice of stepped path for disposal is effective under certain conditions.” I completely disagree with this statement. This paper did not argue or successfully compare any aspects of governance nor model building under different conditions.

Each sub-section of the Introduction and Literature Review sections are poorly organized, without a real beginning, middle, and end. It’s incredibly difficult to determine what the point is to each section because no clear story or summary is being told. The Introduction and Literature Review sections do nothing to set the story for the aspects of a proposed model of waste collection as it seems the author starts to talk about in section 3.

Overall, the writing is wordy and way too much. It needs to be made to be concise and clear. There are many run-on sentences with no clear point or purpose. Further, the figures and tables are poorly explained and as such do not add any understanding to the paper.

Specific Comments by section:

Abstract

The first sentence of the abstract doesn’t make sense. What is the “battlefield” in this sentence?

Introduction

Overall the introduction is poorly organized and constructed. I cannot get the detailed points out of the introduction that I should be able to. The extent to which I understand the purpose is that healthcare waste management is important and underwhelming in China, and that has been highlighted in the current pandemic. I gained no understanding of the extent of this field of study because the referenced works were so poorly and inadequately weaved together. This section requires significant re-writing, both for content and English language.

Line 27-28: What is a designated hospital? What is a square hospital? Please remove "main grip".

Line 29-30: Please correct the name of the IDNDR to the proper name. See https://www.preventionweb.net/organizations/2672#:~:text=An%20International%20Decade%20for%20Natural,developing%20countries%2C%20loss... – Also please cite the organization.

Lines 33-35: Please provide references to these regulations and organizations mentioned

Line 40-44: Improper citation. This reference is unclear as well. It’s hard to tell what the authors are trying to reference out of the work.

Literature Review

Line 96-97: It is unclear what is meant by the idea that green governance has officially entered the historical scene.

Line 131-133: This is an inappropriate citation. “Collaboration” was not first proposed in 2005, plus the paper cited is about the interpretation of consciousness in multiple dimensions.

Line 140-146: Potential plagiarism as a reference is not cited.

Line 229-234: Please eliminate MOST of the use of the word technology from these sentences.

Lines 229-245: Potential plagiarism as there is only 1 reference in the whole paragraph and a lot of things that should have a reference as they are not common knowledge.

Lines 246-261: Potential plagiarism as there are NO references in the whole paragraph and a lot of things that should have a reference as they are not common knowledge.

Line 257-261: Finally, the objective of the paper… not sure why it’s here for the first time. Please make the objective of the paper much clearer and from the beginning. All of the other text should be framed around this objective.

A Model for medical waste disposal based on green governance theory

Line 284-316: Plagiarism as there are NO references in the whole section and a lot of things that should have a reference as they are not common knowledge.

Figure 1: Typically figures help to solidify and enhance the story. This one just further confuses what is already written as none of the items discussed in the Figure have been clearly discussed in the text. Further, it is not clear if this a model in use now, in the literature, or developed by the authors.

Lines 425-436: I am unable to make any sense of these lines.

Figure 3: The map is unclear. Is this a map of Wuhan? If so label it. Secondly, does the blue star indicate that the disposal site is outside of Wuhan because it just looks like an error floating off to the side?  

Line 443: What are optimization results? It seems a bit more important to go into the specifics of the model and methods (i.e., how will optimization be designed) rather than repeating the need to reduce the risk of secondary infection spread, which at this point in the paper has been said at least a dozen times.

Line 451-469: Potential plagiarism as there is only 1 reference in the whole paragraph and a lot of things that should have a reference as they are not common knowledge. None of the algorithms are referenced in use or in theory.

Line 486-489 and line 509: Needs references for data sources.

Line 530: I thought Ant Colony Algorithm was abbreviated ACO but in this line it says AG.

Line 522-558: Prior to this section, the presentation and descriptions of the algorithm types tried are very poor. Although this section's clarity is still not great (and it needs references), it should have come well before the model-building to explain the methods. In fact, a methods section would help this paper immensely. Further, the introduction and literature review should have been more related to the topics necessary for the model development and testing rather than two forms of governance that were over-discussed, as they are rather simple concepts. Describing how these model types are relevant to answer the specific question of medical waste disposal is way more important. Specifically, what is the “pheromone” in the medical waste situation?

Line 603-614: A lot of this is redundant as it has been written nearly identically in sections before.

Line 629-638: This is one VERY long run-on sentence. Please break it up.

Recommendations for the application of green synergistic management of medical waste

Overall, it’s really unclear where these recommendations come from. Nothing prior, in the paper, sets us up for these recommendations. They seem out of the left-field, without any real evidentiary backing.

Future research directions

The last paragraph in this section isn’t really about future work but rather a summary. Please label accordingly.

Author Response

Dear Editors and Reviewers:

Thank you for your letter and for the reviewers’ comments concerning our manuscript entitled “Research on Optimization of Healthcare Waste Management System Based on Green Governance Theory in COVID-19” (ID: ijerph-1148168 ).

Those comments are all valuable and very helpful for revising and improving our paper, as well as the important guiding significance to our researches. We have studied comments carefully and have made correction which we hope meet with approval. The main corrections in the paper and the responds to the reviewer’s comments are as flowing:

Abstract

The first sentence of the abstract doesn’t make sense. What is the “battlefield” in this sentence?

Response: At the beginning of 2020, the COVID-19 epidemic was very serious in China. This is a battlefield in non-war period, and it is a battlefield to protect people's life, health and property safety.

Introduction

Overall the introduction is poorly organized and constructed. I cannot get the detailed points out of the introduction that I should be able to. The extent to which I understand the purpose is that healthcare waste management is important and underwhelming in China, and that has been highlighted in the current pandemic. I gained no understanding of the extent of this field of study because the referenced works were so poorly and inadequately weaved together. This section requires significant re-writing, both for content and English language.

Response: This paper summarized the disposal problems of medical waste in Wuhan during the epidemic period. Combined with the green governance theory, and proposed and optimized the disposal plan of medical waste. We think it can provided guiding suggestions for the disposal of medical waste in China, so as to deal with potential risks and provide effective reference for all regions.

We think that collaborative governance theory and green governance theory are in conflict, and collaborative governance theory has no role in the paper, so we delete this part of the discussion. At the same time, the logic and language of the text are improved.

Line 27-28: What is a designated hospital? What is a square hospital? Please remove "main grip".

Response: Designated hospitals are designated for the treatment of suspected and confirmed cases of COVID-19. The purpose is to expand the capacity of COVID-19 patients and prevent re-transmission within the hospital.

The square hospital in the manuscript is wrong. It should be changed to mobile cabin hospital

Line 29-30: Please correct the name of the IDNDR to the proper name. See https://www.preventionweb.net/organizations/2672#:~:text=An%20International%20Decade%20for%20Natural,developing%20countries%2C%20loss... – Also please cite the organization.

Response: Thank you for your correction.We have changed it to the correct name of IDNDR (International Decade for Natural Disaster Reduction)

Lines 33-35: Please provide references to these regulations and organizations mentioned

Response: Regulations on Medical Waste Management and the National Plan for the Construction of Hazardous Waste are from the Ministry of Ecology and Environment, PRC document.

(http://www.mee.gov.cn/ywgz/fgbz/xzfg/201906/t20190628_707961.shtml)

(http://www.mee.gov.cn/gkml/zj/wj/200910/t20091022_172261.htm)

Line 40-44: Improper citation. This reference is unclear as well. It’s hard to tell what the authors are trying to reference out of the work.

Response: In this part, we want to show that the intelligent medical waste intelligent transport system in Wuhan is using has not played a corresponding role and still needs to be improved

Literature Review

Line 96-97: It is unclear what is meant by the idea that green governance has officially entered the historical scene.

Response: Thank you for your advice.This sentence does not express clearly.We have changed it to people put forward the principles of green governance. What we want to express is that people's concept of ecological environmental protection is getting stronger.This principle should be followed in all aspects of social life. Especially in the treatment of medical waste mentioned in this paper.

Line 131-133: This is an inappropriate citation. “Collaboration” was not first proposed in 2005, plus the paper cited is about the interpretation of consciousness in multiple dimensions.

Response: Thank you for your advice. We think the collaborative governance theory does not conform to the main content of the article. We have deleted the content about collaborative governance theory.

Line 140-146: Potential plagiarism as a reference is not cited.

Response: We believe that this is common sense for those engaged in medical waste management.Different organizations will use different methods.There is no code or system to reference.

Line 229-234: Please eliminate MOST of the use of the word technology from these sentences.

Response: We have referred to the suggestions of other reviewers that collaborative governance theory is redundant.Therefore, the information about collaborative governance theory is removed from this paper.

Lines 229-245: Potential plagiarism as there is only 1 reference in the whole paragraph and a lot of things that should have a reference as they are not common knowledge.

Response: We have referred to the suggestions of other reviewers that collaborative governance theory is redundant.Therefore, the information about collaborative governance theory is removed from this paper.

Lines 246-261: Potential plagiarism as there are NO references in the whole paragraph and a lot of things that should have a reference as they are not common knowledge.

Response: We have referred to the suggestions of other reviewers that collaborative governance theory is redundant.Therefore, the information about collaborative governance theory is removed from this paper.

Line 257-261: Finally, the objective of the paper… not sure why it’s here for the first time. Please make the objective of the paper much clearer and from the beginning. All of the other text should be framed around this objective.

Response: We have referred to the suggestions of other reviewers that collaborative governance theory is redundant.Therefore, the information about collaborative governance theory is removed from this paper.

A Model for medical waste disposal based on green governance theory

Line 284-316: Plagiarism as there are NO references in the whole section and a lot of things that should have a reference as they are not common knowledge.

Response: This part of the data is based on the news during the COVID-19 and data from the Ministry of Ecology and Environment of the PRC.

Figure 1: Typically figures help to solidify and enhance the story. This one just further confuses what is already written as none of the items discussed in the Figure have been clearly discussed in the text. Further, it is not clear if this a model in use now, in the literature, or developed by the authors.

Response: Figure 1 is a model of clinical waste management that we have developed. Including the generation, transportation and treatment of medical waste

Lines 425-436: I am unable to make any sense of these lines.

Response: We simplified the content of this part, and then added it into "3.1 Disposal of medical waste during epidemic period in Wuhan".

Figure 3: The map is unclear. Is this a map of Wuhan? If so label it. Secondly, does the blue star indicate that the disposal site is outside of Wuhan because it just looks like an error floating off to the side?  

Response: Thank you for your advice.We have changed the map so that the districts are more clearly defined. Our previous map showed the scope of Hongshan District, and the blue star logo was outside Hongshan District and within Wuhan City. I'm sorry for the trouble caused to your previous reading.

Line 443: What are optimization results? It seems a bit more important to go into the specifics of the model and methods (i.e., how will optimization be designed) rather than repeating the need to reduce the risk of secondary infection spread, which at this point in the paper has been said at least a dozen times.

Response: I am very sorry that only the results are reflected in the paper. We will upload the detailed algorithm optimization process and results through the attachment.

Line 451-469: Potential plagiarism as there is only 1 reference in the whole paragraph and a lot of things that should have a reference as they are not common knowledge. None of the algorithms are referenced in use or in theory.

Response: We searched for more literature and references to illustrate this paragraph.

Line 486-489 and line 509: Needs references for data sources.

Response: This paragraph of the data is based on the news during the COVID-19 and data from the Ministry of Ecology and Environment of the PRC.

Line 530: I thought Ant Colony Algorithm was abbreviated ACO but in this line it says AG.

Response: Thank you for your carefulness. We have made some changes according to the comments of previous reviewers. Thank you again!

Line 522-558: Prior to this section, the presentation and descriptions of the algorithm types tried are very poor. Although this section's clarity is still not great (and it needs references), it should have come well before the model-building to explain the methods. In fact, a methods section would help this paper immensely. Further, the introduction and literature review should have been more related to the topics necessary for the model development and testing rather than two forms of governance that were over-discussed, as they are rather simple concepts. Describing how these model types are relevant to answer the specific question of medical waste disposal is way more important. Specifically, what is the “pheromone” in the medical waste situation?

Response: Thank you for your advice.We will upload the detailed process of algorithm calculation and optimization in the attachment.

Path points: Community collection points, hospitals, and temporary places are used as path points for the ant colony algorithm.Each pathpoint contains multiple dimensions: longitude, dimension, amount of medical waste, and pheromone.

Pheromone (chromosome code) : the pheromone at each waypoint contains the probability that vehicles from each temporary location will pass through this point to the rest of the region.

Line 603-614: A lot of this is redundant as it has been written nearly identically in sections before.

Response: Thank you for your advice. We have deleted the repeated contents.

Line 629-638: This is one VERY long run-on sentence. Please break it up.

Response: Thank you for your advice. We asked professionals to revise the language of the text.

Recommendations for the application of green synergistic management of medical waste

Overall, it’s really unclear where these recommendations come from. Nothing prior, in the paper, sets us up for these recommendations. They seem out of the left-field, without any real evidentiary backing.

Response: In light of the situation of the Chinese Government, we consider it necessary to make such a proposal.

Future research directions

The last paragraph in this section isn’t really about future work but rather a summary. Please label accordingly.

Response: Thank you for your advice. We have marked this paragraph with a new label.

We would like to express our great appreciation to you and reviewers for comments on our paper. Looking forward to hearing from you.
Thank you and best regards.
Yours sincerely,

Xiaoxue Wang

Corresponding author:
Name: Tianle Liu
E-mail: [email protected]

Round 2

Reviewer 3 Report

The authors attempt to revise the manuscript extensively, however there's one essential section should be improved before considering publication:

In Section 2.1, the authors raise the green governance principles for this study, but it has not well presented the application of these principles throughout the research. The empirical study is disconnected with the theoretical discussion.

Author Response

Dear Editors and Reviewers:

Thank you for your letter and for the reviewers’ comments concerning our manuscript entitled “Research on Optimization of Healthcare Waste Management System Based on Green Governance Theory in COVID-19” (ID: ijerph-1148168 ).

We have made a lot of improvements to the whole article based on your comments and the comments of the fourth reviewer.

We would like to express our great appreciation to you and reviewers for comments on our paper. Looking forward to hearing from you.
Thank you and best regards.
Yours sincerely,

Xiaoxue Wang

Corresponding author:
Name: Tianle Liu
E-mail: [email protected]

Reviewer 4 Report

My comments have been attached in a pdf document. 

Author Response

Dear Editors and Reviewers:

Thank you for your letter and for the reviewers’ comments concerning our manuscript entitled “Research on Optimization of Healthcare Waste Management System Based on Green Governance Theory in COVID-19” (ID: ijerph-1148168 ).

Those comments are all valuable and very helpful for revising and improving our paper, as well as the important guiding significance to our researches. We have studied comments carefully and have made correction which we hope meet with approval. The main corrections in the paper and the responds to the reviewer’s comments are as flowing:

Generally and throughout: Please consider adding pandemic after “the COVID-19”.

Response:

We think your suggestion is very good. So we have added the pandemic after “ the COVID-19”

Generally: In the paragraphs following the presentation of Figure one it would be clearer if it was mentioned at least sometimes that this is “the model” or “our proposed model” or “the model presented in Figure 1”, “in this proposed model”, etc. to make it clear that the details you are going into are items related to the model being proposed. That will make it a lot easier for the reader to follow the details provided in the text and relate them to the Figure.

Generally ad throughout: the Headings and sub-headings are not consistently formatted. Please fix.

Response:

We have made changes in the article as you requested to make it easier for readers to understand.

Generally: It would be really helpful to the reader if you described the algorithms in a theoretical sense and then ALSO in an applied sense. The entire section about the models can be cleaned up and made clearer and more concise. There is a lot of bouncing around. I would suggest introducing all the models in a qualitative sense, then a theoretical sense while tying specifically to the project, and then finally into your specific application of them.

Response:

Thank you very much for your suggestions. We have rewritten sections 3.3.1 in the revised manuscript.

In section 3.3.1, in accordance with your suggestions, we have theoretically described three frequently related algorithms: the genetic algorithm (GA), ant colony algorithm (ACO) and simulated annealing algorithm (SA). To help readers understand our research.

Generally: The focus of this paper is still unclear. At one point it a literature review of the problem of medical waste. A somewhat lengthy and redundant one at that. It could use a lot of tightening up for clarity. At another point it is talking about different optimization models without doing a great job of tying them back to the actual problem at hand. Then, without summarizing those results up very well it jumps into some very off topic recommendations that are not supported or in most cases even mentioned in the previous text.

I see there is supplemental material. It is not referenced anywhere in the manuscript text AND it seems like a complete duplicate of sections of what is presented in the manuscript. Honestly the title of the supplemental page is more relevant than the title on the manuscript.

Response:

Thank you very much for your suggestions. We have rewritten 3.3.2 in the revised manuscript.

In Section 3.3.2, we revised the way of introducing the model. When explaining basic assumptions and some abstract concepts such as pheromone, we combined more with the background of medical waste treatment in Wuhan, so as to avoid the disconnection between model explanation and application.

Abstract

Line 8 and 9: Consider rewording the first sentence to read “controlling the COVID-19 pandemic has made significant and strategic strides”

Response:

Thank you for your advice. We have finished the modification.

Line 10 and 11: Consider rewording the to read “large quantities of healthcare treatment waste has”

Response:

Thank you for your advice. We have finished the modification.

Line 18 and 19: Consider rewording to read “Through the establishment of temporary storage facilities along transit routes”

Response:

Thank you for your advice. We have finished the modification.

Line 21: “thought is necessary” instead of think – tense is off

Response:

Thank you for your advice. We have finished the modification.

Introduction

Line 30: The reference to “square hospital” was not corrected as indicated in the response to reviewers. According to the response it’s supposed to be changed to “mobile cabin hospital” - please update

Response:

I'm really sorry for the change of "Square Hospital". After reading the literature, we have made it clear that it should be "Square Cabin Hospital", so we may have forgotten the change in the subsequent articles.Now the modification has been completed. Thank you for your carefulness

Line 30: with “has” became should be “become”

Response:

Thank you for your advice. We have finished the modification.

Line 46: Please change lackness to “lack”

Response:

Thank you for your advice. We have finished the modification.

Line 82 to 84: Please consider rewording this sentence. It is not clear as it is.

Response:

Thank you for your advice. We've done it to make it more understandable.

Line 84: rather than implementation do you mean “involvement” of relevant hospitals

Response:

The word “involvement” is more apt.

Line 86 to 88: Isn’t it that the lack of coordination between hospitals and relevant departments what resulted in untimely and ineffective disposal of medical waste rather than the other way around?

Response:

Thank you for your advice. We have modified the original sentence to “The lack of coordination between hospitals and related departments has resulted in the failure of timely and effective removal and disposal of medical waste in some hospitals. “

Line 92: Furtherly is not a commonly used word. I would suggest rewording this sentence to start with “Implementation of the algorithm derived and effective transport system will rely on the ….”

Response:

Thank you for your advice. We have changed the English of the sentence

Literature Review Clinical Waste Treatment

Line 275: Please remove the mention of collaborative governance.

Response:

Thank you for your advice. We have finished the modification.

A model for medical waste disposal based on green governance principle

Line 309: “fluctuated” instead since “the critical period” is past tense.

Response:

Thank you for your advice. We have finished the modification.

Line 319 to 320: Please change “the medical waste is carried to a large extent Germs, to the logistics staff to bring a high risk of cross-infection” to “the fact that medical waste is highly contaminated presents significant risk of cross-infection to logistics staff.” – The sentence is not clear as is.

Response:

Thank you for your advice. We have finished the modification.

Line 321: I am unclear how a centralized disposal site is scattered. Maybe instead say that there is currently no centralized disposal site, rather two separate ones across town from one another… or something like that.

Response:

       Thanks. When selecting waste disposal sites, we first inquired the addresses of all hospitals and communities in Wuhan one by one,  and then established the map function between longitude and latitude and the two-dimensional Cartesian coordinate  system. Classical k-means algorithm is introduced to determine the number and location of temporary processing points so  as to obtain preliminary results. However, it is difficult to directly add the three constraints in 3.2.2 based on green governance in the iteration process, because the constraints are difficult to be directly quantified. Therefore, we consider the algorithm location results and establish a buffer zone of 4 square kilometers (which can be regarded as a fault-tolerant zone). The medical treatment points can be moved to any point in the buffer zone to ensure all the constraints in 3.2.2.

       In the newly submitted manuscript, we have added the relevant description in Section 3.2.2.

Line 331: I am unclear what “in order to prevent the illegal use of medical waste, is cumbersome in management” is trying to say. Please simplify this sentence.

Response:

Thank you for your advice. We have simplified the sentence

Lines 343 to 353: This is one sentence… a very long run-on sentence. Please break it up to make it clearer. Plus “medical institutions” was duplicated in the middle of it.

Response:

Thank you for your advice. We have changed the English of the sentence.

Site selection and construction of transit shelters

Line 398-400: I am not really sure what is trying to be said in this sentence. There is no independent clause. There are a bunch of dependent clauses not being applied to anything. Please fix.

Response:

Thank you for your advice. We have finished the modification.

Line 401-408: Please end the sentence on line 406 after the reference. Then begin a new sentence with “this notice…” The last and new sentence needs to be cleaned up at the end. Either end with “should try to meet the following requirements” OR “should try to take the following requirements into account”

Response:

Thank you for your advice. We have finished the modification.

Line 430: The serial numbers are in the Table and the medical center type depicted by shapes/colors in the figure. Please correct this sentence so that it accurately reflects that.

Response:

Thank you for your advice. We have finished the modification.

Figure 2: Please add an “and” to your list and state in what area the map is showing. There are two pictures presented. They are not labeled A and B or 1 and 2. Please do so and expand clearly on their content in the Figure caption. If the black boxes are meant to be readable, they are not. If they are not needed, please remove them. Finally, the numbers on the map that indicate the hospitals in not readable. Please darken and enlarge them.

Response:

Thank you for your suggestion. We have changed the picture according to your suggestion.

Systematic transport of medical waste for transit and temporary storage

Line 446: Please clarify what the “low distribution density” means to that it very clear or change it to “the fact that waste is much more spread out”

Response:

We're not going to use the “low distribution density”. This has been changed in Section 3.2.3.

Lines 449 to 460: Neither of these sentences make sense. Please make the points more concise and clear. It is incredibly hard as a reader to follow when you have point-disposal repeated but applying to different things in a single sentence.

Response:

Thank you for your advice. We have made some modifications

Figure 3: This is very clear. Thank you.

Response:

Thank you for your consideration!

Line 466: Please remove “be”

Response:

Thank you for your advice. We have made some modifications

Lines 467 to 472: This sentence is a bunch of dependent clauses and no independent clause. Further, please use semicolons sparingly and do not capitalize the word that comes after a semi-colon.

Response:

Thank you for your advice. We have made some modifications

Overview of transport route simulation using information integration and big data

Lines 481-484: Please fix the spelling errors in this sentence.

Response:

Thank you for your advice. We have made some modifications

What role does GA have in this sector? Has it been used before? It’s unclear how you jump from chromosomes to transport logistics in utility. If you are the first to apply it in this situation, make that clear and explain why it was selected for this context to begin with, which should be separate for why it was chosen following performance. Based on what is presented in the Table 2. it would be nice to read in the text why/how the features of each model type apply to this logistic problem in this paper so that we can better understand why these three models were chosen for evaluation and why the hybrid model. Maybe consider adding a qualitative assessment to Table 2 of how the features of the models (global convergence, efficiency, and applicability) would look form the hybrid model. Is there any competition between models and features?

Response:

Thanks. As a classic path optimization algorithm, GA has been used in a variety of optimization scenarios. Therefore, we are not the first research to use GA, but we are the first to combine GA and ACO. In this study, the method of combination is: taking ACO as the basic framework, introducing the genetic and mutation ideas in GA into the pheromone of ACO, so that the pheromone has attenuation and mutation behavior. The purpose of the combination is to prevent the algorithm solution from falling into the local optimal value. In the revised manuscript, we have added relevant explanations in lines 1053 to 1057.

Introduction and comparative analysis of GA-ACO model

Lines 518 to 520: Since you didn’t shorten what “temporary storage locations” would be referred to herein please remove all the words in the parentheses.

Response:

Thank you for your advice. We have made some modifications

Line 550 ish: What does S.T stand for?

Response:

       Thanks for your careful review. S.T stands for subject to, it is a common notation in mathematics.

Lines 560 to 564: belongs in the section that introduces the algorithms. Then it would allow for the connection of factors related to waste and logistics to these algorithms to be discussed in this space instead. Right next to the equations where it would make most sense to the reader. Again what is the “pheromone” in this specific situation?

Response:

       Thanks for your valuable advice. In section. 3.3.1, the related algorithms are described theoretically for the purpose of comparison and selection. To introduce the factors related to waste and logistics to the algorithm GA-ACO, we have rewritten the section 3.3.2 that introduces the algorithm. Each of the concepts in Section 3.3.2 is explained in conjunction with the application of waste processing.

As for the meaning of pheromone, It’s essence is to contain the information of route exploration. Specifically, it contains the probability that vehicles from each waste disposal point will go to the next route node according to the current load level. In the newly submitted manuscript, we have added relevant descriptions in lines 1089 to 1100 to make it clear.

Line 580 to 581: this is inconsistent with the previous section that said the primary algorithm is GA modified by ACO.

Response:

       Thanks. In the newly submitted manuscript, we have added more text about how the GA and ACO algorithm combined in section 3.3.2 to make it consistent with the previous section. 

Figure 4: I assume that this flow chart only applied to the ACO. Please indicate that in the figure caption and in the text referencing it. It’s not clear from this section, how these two algorithms were actually integrated. Please add clarity.

Response:

Thanks for your advice. Specifically, the method of combination is: taking ACO as the basic framework, introducing the genetic and mutation ideas in GA into the pheromone of ACO, so that the pheromone has attenuation and mutation behavior. The purpose of the combination is to prevent the algorithm solution from falling into the local optimal value. In the revised manuscript, we have added relevant explanations in lines 927-931 and 1053 to 1057.

Table 3: This is the first mention of Wolf-pack algorithm. Please explain what it is, why it was considered and add it to Table 2 with qualitative descriptions.

Response:

       Thanks for your careful review. After serious discussion, we believe that the wolf colony algorithm and the ant colony algorithm have a certain degree of similarity, and the related experiments have a certain degree of repetition. Therefore, we delete it in the newly submitted manuscript.

Lines 633 to 644: This entire section is a duplicate of lines 513 to 526 in section 3.3.2 Except for the reference to Figure 5.

Response:

Thanks for your careful review. We have a summary here。

Figure 5: Are these the routes derived from the hybrid model since it performed the best? If so, why does the sentence that mentions Figure 5 only mention ACO?  Please make this clear.

Response:

       Thanks for your careful review. We have corrected the sentence that mentions Figure 5 in line 1179. Figure 5 is the result of GA-ACO model. Besides, we have redrawn all the figures to make them clear.

Promote efficient operation optimization management in simulation experimental scenarios

Line 661: What is the “double-link system”?

Response:

The term means that COVID-19 medical waste should be managed separately from common infectious disease medical waste

Line 665: “Ladder”?

Response:

Thanks for your careful review. This word means to manage in a hierarchical way

Line 666 to 669: Please make this sentence more concise. It basically says the same thing twice. Please be consistent with the use of “the COVID-19 epidemic” or “the pneumonia epidemic” throughout the paper. Choose one, do not use both.

Response:

We've changed it to “the COVID-19 epidemic.

Recommendations for the application of green synergistic management of medical waste

Lines 686 to 691: These are all dependent clauses. No independent clause. Please improve this paragraph. Make multiple clear sentences.

Response:

Thank you for your advice. We have made some modifications

Figure 6: This figure is still hard to read. Is this supposed to represent the “Ladder” system? If so, please choose one (ladder or hierarchy)

Response:

Yes, this is the ladder system.

Where do these recommendations come from? Nothing previously in this paper discusses anything related nor supports any of these VERY specific recommendations, not even the literature review in the beginning. They really come out of nowhere. I understand that recommendations and actionable items from scientific work are important, but these are not supported by this scientific work. I would expect, at most, the recommendations to apply to the location and capacity of transfer stations as well as to how many transport mechanisms would be required to handle the waste generated in a timely manner. This paper talks nothing of how to hold collected waste (in bags, etc.) but has related recommendations at the end. Seems very strange.

Response:

We believe that these suggestions are the conclusions we have drawn by combining China's actual situation with scientific theories and methods

Future research directions

It’s not clear how the theory of green governance was actually incorporated into or considered in the GA, ACO, or hybrid model optimization which is why this paper seems like 3 separate papers smashed together.

Response:

       Thanks for your advice. In order to make the introduction and use of the model more closely integrated with the background of medical waste treatment in Wuhan, in the newly submitted manuscript, we rewrite the relevant parts (sections 3.3.1 and 3.3.2), removing the abstract concept introduction, But combined with the actual background.

We would like to express our great appreciation to you and reviewers for comments on our paper. Looking forward to hearing from you.
Thank you and best regards.
Yours sincerely,

Xiaoxue Wang

Corresponding author:
Name: Tianle Liu
E-mail: [email protected]

Round 3

Reviewer 3 Report

The reviewer accepts this manuscript in the present form. 

Author Response

Thank you for your approval concerning our manuscript entitled “Research on Optimization of Healthcare Waste Management System Based on Green Governance Theory in COVID-19” (ID: ijerph-1148168 ).

Reviewer 4 Report

Thank you for making many of the edits that I suggested in the second round.

As displayed below, I think this paper requires more edits before it is ready for publication.

I still feel that across the board the text should be reduced substantially. There is so much, barely relevant, yet redundant information in here that the actual novelty (model optimization methods) is nearly completely missed. If not reduced, it should be made clearer and concise.

Please get rid of the supplemental material. It adds nothing to the paper and is not referenced once.

Generally and throughout:

I realize this has been changed a few times but when referring to COVID-19 either say “the COVID-19 pandemic” or just “COVID-19” not “the COVID-19” or “COVID-19 pandemic” – There are places in the paper that you use the term epidemic instead. Please be consistent. Their definitions are different. Please review.

I brought this up in the previous review, but “transit temporary storage point” is a mouthful. Why is it important to have transit in front of it? These locations are only referred to as temporary storage in Figure 2, why not just shorten it and be consistent? Further, they are called “Temporary Transfer Storage” in Figure 3.

Similarly, in section 3.2.3 the term “medical institution” is used in describing the route. This is a more accurate term to describe the sources of medical waste assessed in this paper, but it is not consistently used throughout. For example, Figure 2 has them all as Hospitals.

This paper is jargon-heavy and wordy. If you would like this to be more broadly applicable across fields and digestible; I would consider reducing the jargon and translating the technical aspects into lay words with a focus on being concise.  

Specific Comments

Line 17: remove “of”

The introduction is much better. Thank you.

Line 288: “Outbreak” is redundant here, a pandemic is a global outbreak.

Line 293: End the sentence after “priority”

Line 363: add “for” between “advocates” and “the”

Line 367: do not capitalize “separation”

Line 448-457: Thank you for adding this section

Figure 2: This is better but the black box is still not readable. Also, please update the text in Line 458-461 to reflect the actual terminology used in the Figure. For example, “shelters” are not mentioned at all in the Figure. Also be clear in the text that these are proposed or existing locations for medical waste treatment, community collection, and temporary storage. Obviously, the hospitals are existing. Be consistent with the terminology used to describe the different aspects. Transit shelters vs. temporary storage, etc. Verify that the consistent terminology you decide on is throughout the paper.

Line 482: Do not capitalize “dense”

Line 492: Place the abbreviation for ant colony algorithm here since it’s the first place that it is mentioned.

Line 492 to 496: I would use the word “to” instead of a hyphen for all of these transfer points being described. It would make reading it much clearer. For example, “community to the community collection point to transit temporary storage point” Or actually, this seems a little out of place here, or at least the mention of ACO is premature in my opinion.

Thank you for adding the descriptions of the different models.

Line 555- 569: For most of this paragraph the abbreviations of GA and ACO were abandoned, please use the abbreviations defined. Further, the first half of the paragraph should be reworded to reflect the fact that due to convergence and applicability the singular GA model would be appropriate, however, because it is easy to combine with other algorithms and was slow to convergence, you chose to combine it with ACO to take advantage of the convergence speed and solving ability to get the best of both worlds. I think this paragraph is important to the entire paper and should be written more clearly and concisely, somewhat as I have outlined here.

Line 582: “Standard”?

Line 637 – 639: This sentence is completely out of place. Maybe move it to the start of the next paragraph and reword it to say something like “Since the proposed optimization combines the idea of gene mutation in GA and concept of pheromone in ACO we will describe what they mean in our study and in mathematical terms” – Did a good job explaining and highlight the concept of pheromone from ACO but the concept of gene mutation from GA is not clearly explained or related to the specific application. Please update.

Line 653: Use the abbreviation; ACO

Line 655: add “, and” before “pheromone”

Line 665: Start the sentence with “Equation 6”

Line 673: Isn’t it waste trucks or drivers in this case, not ants… please change the vocabulary to represent the situation you are studying. Or say something like “ants (or in our case trucks)”

Figure 5: Please update this figure similarly to Figure 2 when changed.

Line 761-763: Why call out this specific location? It’s unclear what this is and why it’s important to mention.

Line 792: “establish” not “establishing”

Line 808-812: Change to: “All closed communities should adopt a management system that places a single person in charge of each building. In principle ….”

Line 826: Need an “and” between “clinics,” and “community”

Line 834: Remove “causing secondary transmission of the virus”

Line 844: temporary is spelled wrong

Line 882: Assume this should be “Medical” not “vedical”

Line 908-909: Drop “what effect and” and edit enrich and refine to “enrichment and refinement of”

The recommendations: Some of these are so wordy and long that it’s hard to tell what the takeaway really is.

Supplemental Material: It is still unclear to me what this adds to the paper. All of its content is already presented in the main body of the paper. Granted, I think that this paper could be much more concise and therefore would reach a larger audience, but some of that feedback has not been taken in the previous two reviews. Still, the supplemental material is not referenced anywhere in the manuscript.

Author Response

Dear Editors and Reviewers:

Thank you for your letter and for the reviewers’ comments concerning our manuscript entitled “Double Path Optimization of Transport of Industrial Hazardous Waste based on Green Supply Chain Management” (ID: sustainability-1167256 ).

Those comments are all valuable and very helpful for revising and improving our paper, as well as the important guiding significance to our researches. We have studied comments carefully and have made correction which we hope meet with approval. The main corrections in the paper and the responds to the reviewer’s comments are as flowing:

Generally and throughout:

I realize this has been changed a few times but when referring to COVID-19 either say “the COVID-19 pandemic” or just “COVID-19” not “the COVID-19” or “COVID-19 pandemic” – There are places in the paper that you use the term epidemic instead. Please be consistent. Their definitions are different. Please review.

Response: Thank you for your advice. We have all changed it to “COVID-19”.

I brought this up in the previous review, but “transit temporary storage point” is a mouthful. Why is it important to have transit in front of it? These locations are only referred to as temporary storage in Figure 2, why not just shorten it and be consistent? Further, they are called “Temporary Transfer Storage” in Figure 3.

Response: Thank you for your advice. Maybe Chinese and English have different ways of saying the same thing. Fig.2 and Fig.3 have been modified. And both of them have the same statement.

Similarly, in section 3.2.3 the term “medical institution” is used in describing the route. This is a more accurate term to describe the sources of medical waste assessed in this paper, but it is not consistently used throughout. For example, Figure 2 has them all as Hospitals.

Response: Thank you for your advice. We have changed all Medical institutions into hospitals.

Specific Comments

Line 17: remove “of”

The introduction is much better. Thank you.

Response: Thank you for your carefulness and earnestness. We have deleted it.

Line 288: “Outbreak” is redundant here, a pandemic is a global outbreak.

Response: The title of Section 2.3 The Last Outbreak has been removed.

Line 293: End the sentence after “priority”

Response: Thank you for your carefulness. We end the sentence after "priority".

Line 363: add “for” between “advocates” and “the”

Response: Thank you for your carefulness. The sixth line of section 3.2.1 has been changed.

Line 367: do not capitalize “separation”

Response: Thank you for your carefulness. The tenth line of section 3.2.1 has been changed.

Line 448-457: Thank you for adding this section

Response: Thank you for your approval.

Figure 2: This is better but the black box is still not readable. Also, please update the text in Line 458-461 to reflect the actual terminology used in the Figure. For example, “shelters” are not mentioned at all in the Figure. Also be clear in the text that these are proposed or existing locations for medical waste treatment, community collection, and temporary storage. Obviously, the hospitals are existing. Be consistent with the terminology used to describe the different aspects. Transit shelters vs. temporary storage, etc. Verify that the consistent terminology you decide on is throughout the paper.

Response: The full text term has been modified to Temporary Storage to maintain the consistency of the full text. Section 3.2.2 adds text at the end to make it clearer whether the location in Figure 2 is proposed or existing.

Line 482: Do not capitalize “dense”

Response: Thank you for your carefulness. The seventh line of section 3.2.3 has been changed.

Line 492: Place the abbreviation for ant colony algorithm here since it’s the first place that it is mentioned.

Response: Thank you for your carefulness. This has been changed to use abbreviations.

Line 492 to 496: I would use the word “to” instead of a hyphen for all of these transfer points being described. It would make reading it much clearer. For example, “community to the community collection point to transit temporary storage point” Or actually, this seems a little out of place here, or at least the mention of ACO is premature in my opinion.

Response: The text of the last part at the top of Figure 3 has been modified to avoid the appearance of ACO. In the last part of the first paragraph of section 3.2.3, the word "-" has been changed to "to", Delete "that is, the transfer temporary storage system of medical waste transport reflects the internal coordination mechanism of medical waste management model”.

Thank you for adding the descriptions of the different models.

Response: Thank you for your approval.

Line 555- 569: For most of this paragraph the abbreviations of GA and ACO were abandoned, please use the abbreviations defined. Further, the first half of the paragraph should be reworded to reflect the fact that due to convergence and applicability the singular GA model would be appropriate, however, because it is easy to combine with other algorithms and was slow to convergence, you chose to combine it with ACO to take advantage of the convergence speed and solving ability to get the best of both worlds. I think this paragraph is important to the entire paper and should be written more clearly and concisely, somewhat as I have outlined here.

Response: Thank you for your advice. The abbreviations GA and ACO have been used in this text. We rewrote the last section of Section 3.3.1 to explain more clearly why the GA-ACO hybrid model was chosen.

Line 582: “Standard”?

Response: In the seventh line of Section 3.3.2, this is the English translation of a Chinese policy on medical waste disposal "Standard for pollution control on medical waste treatment and disposal".

Line 637 – 639: This sentence is completely out of place. Maybe move it to the start of the next paragraph and reword it to say something like “Since the proposed optimization combines the idea of gene mutation in GA and concept of pheromone in ACO we will describe what they mean in our study and in mathematical terms” – Did a good job explaining and highlight the concept of pheromone from ACO but the concept of gene mutation from GA is not clearly explained or related to the specific application. Please update.

Response: Misplaced content has been deleted. In addition, there is an annotation about gene mutation and gene inheritance on page 13.

Line 653: Use the abbreviation; ACO

Response: The ACO is abbreviation.

Line 655: add “, and” before “pheromone”

Response: We have changed it so that formula 6 is in the top two lines.

Line 665: Start the sentence with “Equation 6”

Response: Change "which" under Formula 6 to "Formula (6)".

Line 673: Isn’t it waste trucks or drivers in this case, not ants… please change the vocabulary to represent the situation you are studying. Or say something like “ants (or in our case trucks)”

Response: Thanks for your advice. The content has been rewritten to avoid confusion for the reader.

Figure 5: Please update this figure similarly to Figure 2 when changed.

Response: Thanks for your advice. We have modified the synchronization with Figure 2. The terms maintain the coherence of the text. An example is unified use of “temporary storage”.

Line 761-763: Why call out this specific location? It’s unclear what this is and why it’s important to mention.

Response: Thanks for your advice. We have delete the word "At" At the beginning of the second paragraph of Section 3.3.4.

Line 792: “establish” not “establishing”

Response: Thanks for your advice. The first line of Part IV has been changed.

Line 808-812: Change to: “All closed communities should adopt a management system that places a single person in charge of each building. In principle ….”

Response: Thanks for your advice. We have finished the modification.

Line 826: Need an “and” between “clinics,” and “community”

Response: Thanks for your advice. We have finished the modification.

Line 834: Remove “causing secondary transmission of the virus”

Response: Thanks for your advice. We have finished the modification.

Line 844: temporary is spelled wrong

Response: Thanks for your advice. We have finished the modification.

Line 882: Assume this should be “Medical” not “vedical”

Response: Thanks for your advice. We have finished the modification.

Line 908-909: Drop “what effect and” and edit enrich and refine to “enrichment and refinement of”

Response: Thanks for your advice. We have finished the modification.

The recommendations: Some of these are so wordy and long that it’s hard to tell what the takeaway really is.

Response: Thanks for your advice. We have finished the modification.

Supplemental Material: It is still unclear to me what this adds to the paper. All of its content is already presented in the main body of the paper. Granted, I think that this paper could be much more concise and therefore would reach a larger audience, but some of that feedback has not been taken in the previous two reviews. Still, the supplemental material is not referenced anywhere in the manuscript.

Response: Thanks for your advice. We have finished the modification.

We would like to express our great appreciation to you and reviewers for comments on our paper. Looking forward to hearing from you.
Thank you and best regards.
Yours sincerely,

Xiao-xue Wang

Corresponding author:
Name: Tianle Liu
E-mail: [email protected]
